# Smart "*Chef*": Verifying the Effect of Role-based Paraphrasing for Aspect Term Extraction

**Jiaxiang Chen, Yu Hong**[*]**, Qingting Xu, Jianmin Yao**
School of Computer Science and Technology, Soochow University, SuZhou, China
{xxfz56, tianxianer, qtxu0801}@gmail.com, jyao@suda.edu.com

## Abstract

We tackle Aspect Term Extraction (ATE), a task of automatically extracting aspect terms from sentences. The current Pretrained Language Model (PLM) based extractors have achieved significant improvements. They primarily benefit from context-aware encoding. However, a considerable number of sentences in ATE corpora contain uninformative or low-quality contexts. Such sentences frequently act as "troublemakers" during test. In this study, we explore the context-oriented quality improvement method. Specifically, we propose to automatically rewrite the sentences from the perspectives of virtual experts with different roles, such as a "*chef*" in the restaurant domain. On this basis, we perform ATE over the paraphrased sentences during test, using the well-trained extractors without any change. In the experiments, we leverage ChatGPT to determine virtual experts in the considered domains, and induce ChatGPT to generate paraphrases conditioned on the roles of virtual experts. We experiment on the benchmark SemEval datasets, including *Laptop*-domain L14 and *Restaurant*-domain R14-16. The experimental results show that our approach effectively recalls the inconspicuous aspect terms like "*al di la*", although it reduces the precision. In addition, it is proven that our approach can be substantially improved by redundancy elimination and multi-role voting. More importantly, our approach can be used to expand the predictions obtained on the original sentences. This yields state-of-the-art performance (i.e., $F1$-scores of 86.2%, 89.3%, 77.7%, 82.7% on L14 and R14-16) without retraining or fine-tuning the baseline extractors.

## 1 Introduction

ATE is a natural language processing task, which aims to extract aspect terms from sentences ([Jakob and Gurevych, 2010](#)). The *aspect term* refers to a word, phrase or named entity depicting a certain

---
[*]Corresponding author.

domain-specific attribute. For example, the text span "*al di la*" in (1) is specified as a *restaurant*-domain aspect term because it appears as the sign of an Italian trattoria.

(1) *Love* al di la (Selected from SemEval-R15).

(2) *We take pride in every dish we serve at* al di la (Rewritten by ChatGPT with a role of "*chef*").

The current studies leverage PLMs as backbones to construct ATE models (extractors), including BERT ([Devlin et al., 2019](#)), BART ([Lewis et al., 2020](#)) and T5 ([Raffel et al., 2020](#)) as mentioned in Section 2. They yield significant improvements, compared to conventional neural networks like CNN ([LeCun et al., 1998](#)). The advantage is primarily attributed to the strong perception ability on noteworthy contexts, as well as context-aware representation learning ability.

However, such extractors frequently suffer from uninformative or low-quality contexts. For example, the context "*Love*" in (1) is uninformative for recognizing "*al di la*". By contrast, the substitution containing a knowledge-rich context makes it easier to recognize aspect terms, such as the case in (2). Accordingly, we propose a ChatGPT-based Edition Fictionalization (CHEF) method to assist the current PLM-based extractors. CHEF acts as a domain-specific virtual expert with different roles to rewrite sentences, with the aim to refine contexts of potential aspect terms. ChatGPT is utilized for both expert generation and sentence rewriting. A series of post-processing methods are coupled with CHEF, including redundancy elimination, synonym replacement and multi-role voting.

We experiment on the benchmark datasets L14 and R14-16 ([Pontiki et al., 2014](#), [2015](#), [2016](#)). The well-trained $BERT_{base}$ and PST ([Wang et al., 2021](#)) (SoTA) are adopted in the experiments. During test, they perform over the rewritten sentences by CHEF, without retraining and fine-tuning. The test results

demonstrate the effectiveness of CHEF in recalling aspects and expanding the predictions.

## 2 Related Work

Context-aware encoding contributes to ATE. It brings domain-specific contextual information into token-level representations. Therefore, CNN (LeCun et al., 1998; Karimi et al., 2021) and BERT (Devlin et al., 2019; Karimi et al., 2021; Klein et al., 2022) are widely used for ATE due to the abilities of convolving local contexts or absorbing attentive contextual information. Their expanded versions DECNN (Xu et al., 2018; Wei et al., 2020; Li et al., 2020; Chen and Qian, 2020) and BERT-PT (Xu et al., 2019; Wang et al., 2021; Chen et al., 2022b) are generally adopted as backbones in the subsequent studies. In addition, BERT is utilized as the pedestal for context-aware encoding in a series of more complex tasks, including Aspect-Sentiment Triplet Extraction (ASTE) (Chen et al., 2022a; Zhang et al., 2022b; Chen et al., 2022c; Zhang et al., 2022a; Chen et al., 2022d; Zhao et al., 2022b) and MRC-based ASTE (Yang and Zhao, 2022; Zhai et al., 2022). Recently, the generative framework is introduced into the studies of ASTE, and accordingly BART (Yan et al., 2021; Zhao et al., 2022a) and T5 (Zhang et al., 2021; Hu et al., 2022) are used. They are constructed with the transformer encoder-decoder architecture in the seq2seq paradigm, where attentive contextual information absorption is conducted on both sides.

## 3 Approach

We aim to provide knowledge-rich and high-quality sentences for ATE. Specifically, we use CHEF to rewrite sentences, and feed the rewritten sentences into an extractor to predict aspect terms. Finally, we combine the aspect terms which are respectively extracted from the original and rewritten sentences.

### 3.1 Extractors

We follow the common practice (Chernyshevich, 2014; Toh and Wang, 2014; San Vicente et al., 2015) to treat ATE as a sequence labeling task. B/I/O labels are used, which respectively signal **B**eginning, **I**nside and **O**utside tokens relative to aspect terms. Therefore, an extractor essentially classifies each token into one of the B/I/O labels in terms of the token-level hidden state. We use PLM to compute the hidden states of tokens, and use a Fully-connected (FC) layer for classification.

| Domain | Virtual Experts |
|---|---|
| Restaurant | Restaurant Owner, Chef, Waiter, Diner, Catering Consultant, Receptionist, Cleaning Staff, Purchasing Manager, Restaurant Manager, Financial Officer |
| Laptop | Hardware Engineer, Software Engineer, Technical Support Engineer, Marketer, Customer Service Staff, Test Engineer Case Designer, Supply Chain Manager, Purchasing Manager, User |

Table 1: Experts of *Restaurant* and *Laptop* domains.

---

**Algorithm 1:** Rewrite with Zero-shot prompting

**Input:** Test set $\mathcal{S}$, Role set $\mathcal{R}$ **AND** Domain set $\mathcal{D}$
**Output:** All the rewritten sentences
**Prompt**.fmt($\mathcal{D}_i$,$\mathcal{R}_j$,$\mathcal{S}_k$) = **Rewrite** $\mathcal{S}_k \in \mathcal{S}$ from the perspective of $\mathcal{R}_j \in \mathcal{R}$ in the domain of $\mathcal{D}_i \in \mathcal{D}$
**foreach** $\mathcal{D}_i \in \mathcal{D}$ **do**
    **foreach** $\mathcal{R}_j \in \mathcal{R}$ **do**
        **foreach** $\mathcal{S}_k \in \mathcal{S}$ **do**
            *Instruction* = **Prompt**.fmt ($\mathcal{D}_i$,$\mathcal{R}_j$,$\mathcal{S}_k$)
            *Prediction* = **ChatGPT**(*Instruction*)
        **end**
    **end**
**end**

---

We consider two PLMs for hidden-state computation in the experiments, including $\text{BERT}_{base}$ and $\text{BERT}_{pt}$-based PST (Wang et al., 2021).

### 3.2 CHEF

CHEF comprises two stages, including role generation of domain-specific virtual experts, as well as role-based rewriting. It is coupled with three post-processors, including redundancy elimination, synonym replacement and multi-role voting.

**Role Generation**– CHEF induces ChatGPT[1] to generate a series of virtual experts playing different roles. The generation is prompted by the target-domain name $\mathcal{D}_i$ like "*Restaurant*". The query we use is as follows: "*Output the roles of experts in the domain of [$\mathcal{D}_i$] according to the different responsibilities.*". Table 1 shows the experts.

**Sentence Rewriting**– Given a sentence $\mathcal{S}_k$ in the ATE datasets, CHEF induces ChatGPT to rewrite the sentence from the perspective of a role-specific expert $\mathcal{R}_j$. Zero-shot prompting (Wei et al., 2022) is used during generation. In other words, there isn't any example provided for prompting ChatGPT. The query we use is as follows: "*Rewrite the sentence [$\mathcal{S}_k$] from the perspective of [$\mathcal{R}_j$] in the domain of [$\mathcal{D}_i$].*". We rewrite all the instances in the test sets using Algorithm 1.

**Post-Processing**– We drive the extractors to pre-

---

[1]https://platform.openai.com/playground?mode=chat

| | L-14 | | | R-14 | | | R-15 | | | R-16 | | |
|---|---|---|---|---|---|---|---|---|---|---|---|---|
| | Train | Dev | Test | Train | Dev | Test | Train | Dev | Test | Train | Dev | Test |
| **#Sentence** | 2,436 | 609 | 800 | 2,433 | 608 | 800 | 1,052 | 263 | 685 | 1,600 | 400 | 676 |
| **#Aspect** | 1,940 | 418 | 654 | 2,950 | 744 | 1,134 | 948 | 252 | 542 | 1,408 | 336 | 612 |

Table 2: Statistics of ATE datasets. #Sentence and #Aspect denote the number of sentences and aspects.

dict aspect terms over the role-specific rewritten sentences. Redundant results may obtained, which are specified as the aspect terms never occurring in the original sentences. For example, although the token "*dish*" in the rewritten sentence in (2) is correctly predicted as an aspect term, it is redundant due to non-occurrence in the original sentence in (1). We filter the redundant results during test.

In addition, we use a soft synonym replacement method to reduce false positive rates. Specifically, given an original sentence $S_i$ and the rewritten case $\mathring{S}_i$, we segment both of them into $n$-grams. Assume a set $\mathcal{U}_i$ of $n$-grams ($1 \leq n \leq 5$) in $S_i$ share a part of the predicted aspect term with the set $\mathring{\mathcal{U}}_i$ of $n$-grams in $\mathring{S}_i$, we calculate the similarity between each gram $u_{ij}$ in $\mathcal{U}_i$ and every gram $\mathring{u}_{ik}$ in $\mathring{\mathcal{U}}_i$. We rank all pairs of $\{u_{ij}, \mathring{u}_{ik}\}$ in terms of their similarities, and select the top-1 ranked $n$-gram pair for synonym replacement, i.e., $\mathring{u}_{ik} \Leftarrow u_{ij}$. Meanwhile, the replaced $n$-gram is specified as the unabridged aspect term, as shown in the example in (3). Note that Cosine similarity is computed over the embeddings of each pair $\{u_{ij}, \mathring{u}_{ik}\}$. The embedding of each $n$-gram is obtained by conducting mean pooling (Reimers and Gurevych, 2019) over the token-level hidden states.

(3) **Original**: *Best Indian food I have ever eaten.*
 **Rewritten**: *I put a lot of effort into perfecting our Indian dishes* [Aspect Term]. (Rewritten by CHEF with a role of "*chef*")
 **Replacement**: *Indian dishes* $\Leftarrow$ *Indian food*
 **Output**: *Indian food*

**Multi-role Voting**– We conduct multi-role voting only if an extractor obtains controversial results from the original and rewritten sentences. It facilitates the combination of the extraction results.

Assume an extractor refuses to extract a text span $t_i$ as an aspect from the original sentence, though it would like to do so from the sentences rewritten by different roles of experts, thus we regard $t_i$ as a controversial result. In this case, we define the behavior of extracting $t_i$ as the voting for acceptance, otherwise rejection. On this basis, we compute the acceptance rate $v_i$ over the rewritten

sentences of all experts: $v_i = N_c/N_{all}$, where $N_c$ denotes the number of voting for acceptance, while $N_{all}$ is the number of experts. $N_{all}$ is set to 10 in our experiments. For the *Restaurant* domain, a controversial result $t_i$ is finally adopted only if $v_i$ is no less than a threshold of 0.7. For the *Laptop* domain, the threshold is set to 0.8.

## 4 Experimentation

### 4.1 Datasets and Evaluation

We experiment on the SemEval datasets, including L14 and R14-16 (Pontiki et al., 2014, 2015, 2016). All the instances in L14 are selected from the *Laptop* domain, while those in R14-16 derive from the *Restaurant* domain. We follow the common practice (Dai and Song, 2019; Wang et al., 2021) to split the datasets into training, validation and test sets. The statistics in them are shown in Table 2.

It is noteworthy that only the extractors are trained and developed using the above datasets. CHEF has nothing to do with training and development. It is conducted only on the test sets, providing rewritten sentences for the extractors and post-processes the predictions. We evaluate all the models using $F1$-score (Chernyshevich, 2014).

### 4.2 Hyperparameters

We respectively use $\text{BERT}_{base}$ and $\text{BERT}_{pt}$-based PST (Wang et al., 2021) to construct the extractors, where PST achieved the best performance so far.

For $\text{BERT}_{base}$, we set the maximum sequence length to 128 and carry out training in 4 epochs. The optimization of model parameters is obtained using AdamW with a learning rate of 3e-5 and a weight decay of 0.01. We set the batch size to 10. For PST, we adopt their initial hyperparameters, setting the first round of training to 5 epochs and performing 4 rounds of self-training. The learning rate is set to 5e-5. AdamW is used as the optimizer. All the other hyperparameters remain consistent with the reported ones.

### 4.3 Comparison Result

In Table 3, we report the performance of PST which is enhanced by CHEF, along with other state-of-

| Model | L14 | R14 | R15 | R16 |
|---|---|---|---|---|
| ChatGPT (2023) | 43.03 | 55.65 | 40.33 | - |
| +5-shot ICL (2023) | 48.19 | 70.99 | 53.49 | - |
| +5-shot COT (2023) | 54.50 | 72.41 | 59.27 | - |
| DECNN (2018) | 81.59 | - | - | 74.37 |
| +CDA (2020) | 81.58 | - | - | 75.19 |
| +Repositioning (2020) | 84.17 | 84.55 | 72.03 | 75.40 |
| +PrototypeE (2020) | 83.19 | 87.39 | 73.27 | 76.98 |
| BERT$_{base}$ (2019) | 79.86 | 86.58 | 68.08 | 73.50 |
| +Repositioning (2020) | 81.43 | 87.10 | 72.68 | 77.71 |
| +CDA (2020) | 81.14 | - | - | 75.89 |
| +PST (2021) | 84.17 | 87.63 | 72.81 | 77.09 |
| BERT$_{pt}$ (2019) | 84.37 | 88.41 | 73.66 | 78.29 |
| +CDA (2020) | 85.33 | - | - | 80.29 |
| +PST (2021) | **86.91** | 88.75 | 75.82 | 82.56 |
| Ours (PST+CHEF) | 86.22 | **89.33** | **77.69** | **82.74** |

Table 3: The performance (F1-score) of various methods. "ICL" refers to in-context learning, while "COT" stands for chain-of-thought. The sign "+" represents a certain method combined with the baseline model.

| Model | L14 | R14 | R15 | R16 |
|---|---|---|---|---|
| BERT$_{base}$ | 82.96 | 88.03 | 72.17 | 74.88 |
| +CHEF | 83.39 | 88.25 | 72.92 | 75.56 |
| PST (Reproduced) | 86.13 | 89.28 | 77.29 | 82.62 |
| +CHEF | 86.22 | 89.33 | 77.69 | 82.74 |

Table 4: Improvements (F1) on different extractors.

the-art ATE models. In this case, the extraction results are obtained by combining the predictions of PST on both original and rewritten sentences, where multi-role voting is used. It can be found that CHEF enables PST to achieve better performance, slightly increasing the performance gap relative to other strong competitors.

In Table 4, we report the effects of CHEF on both the BERT$_{base}$ and BERT$_{pt}$-based PST, where multi-role voting is used for prediction combination. It can be observed that CHEF enables both extractors to achieve better performance on all the test sets, without retraining and fine-tuning.

### 4.4 Discussion

In two separate experiments, we demonstrate that CHEF contributes to the salvage of the missed aspect terms, and yields more substantial improvements on short or long sentences.

**Salvage Rate–** We select all the incompletely-solved sentences from the test sets, each of which contains at least one aspect term neglected by the extractor. On this basis, we use CHEF to rewrite the sentences and drive the extractor to rework them. In this experiment, we consider the BERT$_{base}$ extractor, and verify the changes in recall rates. Figure

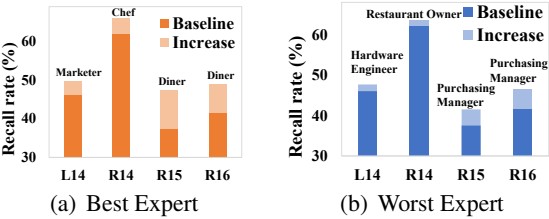

(a) Best Expert    (b) Worst Expert

Figure 1: Significant improvements of *Recall* rates yielded by CHEF on the incompletely-solved sentences.

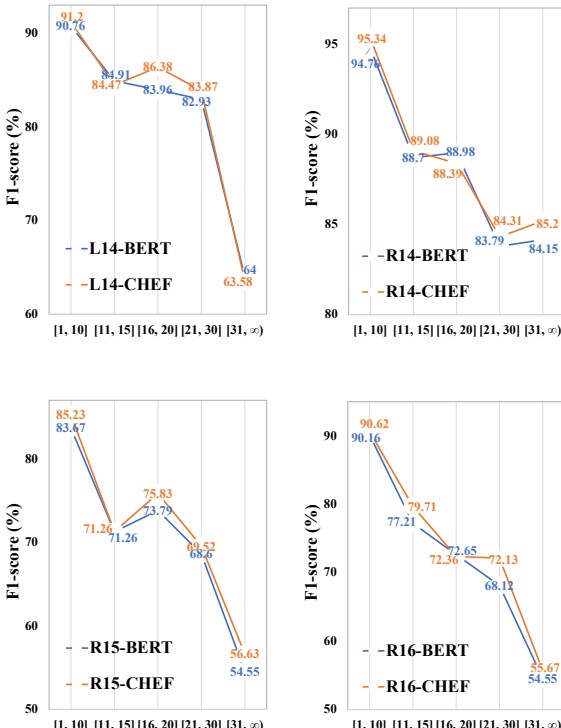

Figure 2: F1-scores for the sentences at different lengths. Appendix A gives the statistics about them.

1 shows the experimental results, where only the performances yielded by the best and worst experts are provided. It can be observed that CHEF substantially improves the recall rates for all test sets, no matter whether it plays the role of best expert or worst. The most significant improvement in recall rate reaches about 9%. Besides, the change of precision, recall and F1 score on the full test set can be found in Appendix B.

**Adaptability–** CHEF applies more to short and long sentences. The former generally contains uninformative contexts, while the latter contains lower-quality contexts due to noises. We split each test set into different subsets according to the lengths of sentences. There are five subsets obtained for each,

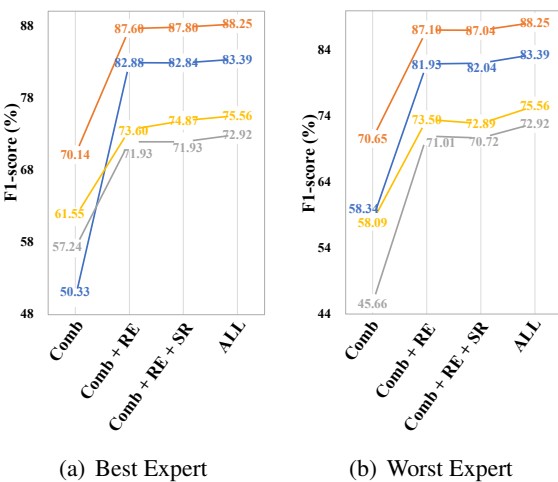

|                | (a) Best Expert | (b) Worst Expert |
|----------------|-----------------|------------------|

Figure 3: Results of ablation experiments ($F$1-scores).

including the sentences having a length within the ranges of $[1, 10], [11, 15], [16, 20], [21, 30]$ and no less than 30. The statistics in the subsets are shown in Table 5. Figure 2 shows the ATE performance at the original sentences of different lengths, and that at the rewritten cases by CHEF (without multi-role prediction combination). It can be observed that CHEF only yields improvements for relatively short or long sentences. The most significant improvement reaches about 4% $F$1-score at R16.

## 4.5 Ablation Study

CHEF consists of four components, including **Comb**ination (Comb), **R**edundancy **E**limination (RE), **S**ynonym **R**eplacement (SR) and Multi-role Voting, as presented in Section 3. To verify the effects of the components, we conduct an ablation experiment. Figure 3 illustrates the verification results over best and worst experts, where ALL indicates the complete CHEF method.

It can be observed that the simple combination (Comb) causes significant performance degradation, although many terms the baseline missed are salvaged. When redundancy elimination (RE) is used, the comparable (a little worse) performance to the baseline is achieved. At this time, the precision is still lower because the synonymous terms are regarded as negative examples during evaluation. When synonym replacement (SR) is used, the performance is increased for some roles while not for others. When multi-role voting is used, the disagreement among the roles is resolved, and thus the performance is increased to a relatively higher level. We provide examples in Appendix C to facilitate the understanding of our ablation study.

## 5 Conclusion

We utilize ChatGPT to rewrite sentences with different roles of domain-specific experts, so as to provide informative and high-quality contexts for PLM-based ATE models. Experiments show that the proposed method contributes to the salvage of the neglected aspect terms, and applies more to the short and long sentences. In the future, we will use the rewritten sentences for contrastive learning. To reduce the reliance on ChatGPT, we will develop an offline context rewriting method by knowledge distillation and domain-specific pretraining.

## Limitations

We propose to use ChatGPT as an auxiliary toolkit to produce informative and high-quality contexts for context-aware token-level encoding, so as to enhance PLM encoders for aspect term recognition. Our experiments show that the proposed method yields slight improvements when coupled with strong domain-specific models, and it is not only effective in recalling neglected cases, but performs better for short and long instances. Unavoidably, the proposed method has the limitations in building a self-contained model, due to the lack of training and fine-tuning. To overcome the problem, we will develop a lite comparable generator to ChatGPT by knowledge distillation and domain-specific pretraining. Furthermore, we will incorporate role-based rewritten sentences into the training process, with the support of contrastive learning.

## Acknowledgment

The research is supported by National Key R&D Program of China (2020YFB1313601) and National Science Foundation of China (62376182, 62076174).

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

| | [1, 10] | | [11, 15] | | [16, 20] | | [21, 30] | | > 30 | |
|---|---|---|---|---|---|---|---|---|---|---|
| | #Sentence | #Aspect | #Sentence | #Aspect | #Sentence | #Aspect | #Sentence | #Aspect | #Sentence | #Aspect |
| L14 | 260 | 126 | 224 | 168 | 156 | 158 | 115 | 122 | 45 | 80 |
| R14 | 218 | 190 | 238 | 287 | 151 | 241 | 137 | 253 | 56 | 163 |
| R15 | 254 | 153 | 148 | 130 | 111 | 108 | 103 | 116 | 69 | 90 |
| R16 | 240 | 161 | 149 | 138 | 125 | 130 | 111 | 120 | 50 | 101 |

Table 5: Statistics of ATE test sets split according to the lengths of sentences.

| | L14 | R14 | R15 | R16 |
|---|---|---|---|---|
| P | 85.97→84.98 | 88.23→86.82 | 70.28→70.07 | 74.76→74.37 |
| R | 80.15→81.85 | 88.85→89.73 | 74.17→76.01 | 75.00→76.80 |
| $F1$ | 82.96→83.39 | 88.03→88.25 | 72.17→72.92 | 74.88→75.56 |

Table 6: Performance changes of Precision, Recall and $F1$-score when applying our method.

| Examples |
|---|
| **Example of redundancy elimination:** |
| **Original:** Better than nobu, tomoe, taka, yama, sugiyama, and the rest. |
| **Waiter:** Our *chef's sushi* is so amazing, it's better than Nobu, Tomoe, Taka, Yama, Sugiyama, and other well-known sushi spots in the city. |
| **Ground-truth:** *[No Aspect]* |
| **Output from the rewritten sentence:** chef's sushi |
| **Example of synonym replacement:** |
| **Original:** Best *Indian food* I have ever eaten. |
| **Chef:** I put a lot of effort into perfecting our *Indian dishes*. |
| **Ground-truth:** Indian food |
| **Output from the rewritten sentence:** Indian dishes |
| **Example of multi-role voting:** |
| **Original:** I have to say they have one of the fastest *delivery times* in the city. |
| **Chef:** I can prepare the order quickly to ensure timely *delivery*. |
| **Restaurant Owner:** Our *delivery times* are some of the fastest in the city. |
| **Catering Consultant:** Their *delivery times* are impressive and contribute to a positive customer experience. |
| **Ground-truth:** delivery times |
| **Output from the Chef:** delivery |

Table 7: Examples of how each component works.

## A Test Set Reorganization

We divide the test sets into different subsets according to the lengths of sentences. For each test set, we obtained five subsets, which can be found in Table 5.

## B Change of Precision and Recall

We provide additional results reflecting the change in precision and recall scores, as shown in Table 6. It can be found that the precision score decreases, meanwhile, the recall score increases. The performance gain at the recall score is because of the enhancement from our post-processing methods.

## C Examples for Ablation Study

Our method combines the extracted results from both the original and rewritten sentences. However, the combination without post-processing causes terrible performance because of noises. To solve the problem, we have introduced the post-processing method, including redundancy elimination, synonym replacement, and multi-role voting. We provide examples that have been post-processed using the above solutions in Table 7.