# OpenReview forum: "Smart “Chef”: Verifying the Effect of Role-based Paraphrasing for Aspect Term Extraction"
_EMNLP/2023/Conference — EMNLP 2023 Findings_

### Official Review · Reviewer_FoJG · 2023-08-05

**Typos Grammar Style And Presentation Improvements:** 039
**Soundness:** 3

**Excitement:**

3: Ambivalent: It has merits (e.g., it reports state-of-the-art results, the idea is nice), but there are key weaknesses (e.g., it describes incremental work), and it can significantly benefit from another round of revision. However, I won't object to accepting it if my co-reviewers champion it.

**Missing References:**

None

**Paper Topic And Main Contributions:**

This paper presents a system which finds sequences in sentences that are “aspects”, which is a first step in aspect-based sentiment analysis (ABSA).  The innovation consists in using ChatGPT to generate new data.  This is not done simply by generating more training data, but by generating loose paraphrases of an input sentence from the point of view of other participants in the domain, and then aligning aspects in the new paraphrases with the original sentence.  This approach is interesting and original.

**Questions For The Authors:**

A) Is ATE useful for something other than ABSA?  If no, have you done experiments comparing with end-to-end systems (presumably generative, such as Zhang et al. 2021)?  This would obviously require integration with an ABSA system which takes the aspect terms as input.

B) Do you have any data that addresses the question which of the components of your approach actually help?

**Reasons To Accept:**

* The paper presents a novel approach to using ChatGPT for improving performance on an NLP problem, which could be of direct relevance to, or of inspiration to, other researchers, not only those interested in ABSA.
* Overall clearly written.
* Great acronym (just kidding).

**Reasons To Reject:**

* The way of using ChatGPT is complex (and interesting), but the authors do not present an ablation study to show which steps help.  Update after rebuttal:  the authors provided a table with ablation study data in the rebuttal (thanks!), but the amount of data is overwhelming without a proper interpretation and integration into the paper.  However, I thank the authors for being responsive to the suggestion that several reviewers had.
* The improvements are small, and in fact smaller than the change in results due to a re-implementation (as confirmed by the authors in rebuttal).  While it is not the standard in our field to report statistical significance, when improvements are really small, it becomes perhaps more urgent to do so.
* If, as this reviewer believes, aspect Term Extraction (ATE) is only useful as a pre-processing step for ABSA, then the reader would also like to see a comparison with end-to-end systems.

**Reproducibility:**

4: Could mostly reproduce the results, but there may be some variation because of sample variance or minor variations in their interpretation of the protocol or method.

**Reviewer Confidence:**

4: Quite sure. I tried to check the important points carefully. It's unlikely, though conceivable, that I missed something that should affect my ratings.

---

> ### Author Rebuttal · Authors · 2023-08-28
>
> We wholeheartedly appreciate your dedicated time in reviewing our paper and offering valuable suggestions to enhance its quality. Your feedback holds immense value for us, and we approach it with utmost seriousness. Each suggestion you provided has been thoroughly considered, and we provide our responses as follows:
>
>
>
> ----------Response to Comments----------
>
>
> **Comment #1:** The way of using ChatGPT is complex (and interesting), but the authors do not present an ablation study to show which steps help.
>
>
> **Response to #1:** We totally agree with your viewpoint. Owing to the page limit, we have not provided a comprehensive list of results. In the extended version, we are enthusiastic about sharing more extensive experimental data. In order to provide a quick insight for reviewers, we show the available data in Table 1 behind our replies, and provide examples (behind Table 1) to assist the analysis. Table 1 exhibits the performance of our approach when different experts are respectively used for paraphrase generation and result combination, as well as the effect of utilizing synonym replacement and redundancy elimination.
>
>
> **Comment #2:** The improvements are small, and in fact smaller than the change in results due to a re-implementation (if I understand correctly). While it is not the standard in our field to report statistical significance, when improvements are really small, it becomes perhaps more urgent to do so.
>
>
> **Response to #2:** Our efforts primarily focused on qualifying the input sentences at the inference stage. This resulted in a slight performance enhancement across various models. The improvements were obtained without changing the previously-trained ATE model, or retraining and fine-tuning it. It preliminarily proved our hypothesis that role-based sentence rewriting can impose positive effects on ATE. Based on the findings and experiments in our submission, there are most probably a variety of interesting and promising methods that can be soon proposed in the future, such as role-based data augmentation, role-based distracting data generation for generative adversarial mechanism, or attention remodeling during training.
>
> In the Appendix behind our replies, we present the statistical significance analysis results, where P-value is used. We hope it can provide deeper insight into the significance. We intend to provide all the results in the extended revision of this paper.
>
> **Comment #3:** If, as this reviewer believes, aspect Term Extraction (ATE) is only useful as a pre-processing step for ABSA, then the reader would also like to see a comparison with end-to-end systems.
>
>
> **Response to #3:** We are actively implementing your recommendations. Specifically, we are in the process of employing ChatGPT to rewrite sentences for the partial training set. Using such data, we are training LLaMA by distillation. The goal is to produce a more efficient and cost-effective rewriter. Our next objective is to validate whether incorporating this data into the training process further improves performance. This will enable us to conduct more comprehensive experiments using end-to-end models.
>
>
>
> ----------Response to Questions----------
>
>
> **Question #1:** Is ATE useful for something other than ABSA? If no, have you done experiments comparing with end-to-end systems (presumably generative, such as Zhang et al. 2021)? This would obviously require integration with an ABSA system which takes the aspect terms as input.
>
>
> **Response to #1:** We are conducting more complete experiments on the ATE task to more fully apply it to tasks such as ABSA and Comparative Opinion Quintuple Extraction (COQE).
>
>
> **Question #2:** Do you have any data that addresses the question which of the components of your approach actually help?
>
>
> **Response to #2:** We are enthusiastic about sharing these experimental data with you (Please see Table 1 and Examples in the Appendix at the end of the response).
>
>
>
> ----------Response to Typos Grammar Style And Presentation Improvements----------
>
>
> We greatly appreciate your valuable suggestions regarding our writing, and we are committed to incorporating them into the revised version of the paper to enhance its overall quality.
>
>
> Besides, we have meticulously considered each of your suggestions in the “Typos Grammar Style And Presentation Improvements”. We answer the concerns as below:
>
>
> **Suggestion #1:** Is this task performed only in the context of sentiment analysis, or is it also important for other tasks? Maybe make this clear from the beginning. (line 039)
>
>
> **Response to #1:** ATE is a subtask of ABSA, and the dataset employed in our study can analyze only ABSA or exclusively focus on ATE. We fully concur with your suggestion, if space permits, and intend to add the introduction at the beginning of the paper.
>
>
> **Suggestion #2:** Why? In the context of sentiment analysis, whatever follows “love” would be as aspect, no? (line 062)
>
>
> **Response to #2:** This is not an absolute situation. In the Res14 training set, we identified 46 instances of the term "Love," where 27 occurrences of noun phrases following "Love" were not annotated as aspect terms. For example, "I love the fact that the pizza tastes so good and is so cheap." In this example, the aspect term is "pizza tastes," rather than "fact".
>
>
> **Suggestion #3:** I don’t understand this. Explain. What is “content”? (line 153)
>
>
> **Response to #3:** We were trying to explain that [S_k] is the content of the original sentence. Our prompt is “Rewrite the sentence [S_k] from the perspective of [R_j] in the domain of [D_i]”, where [S_k] will be replaced by a real sentence (i.e., original sentence) before the prompt is fed into ChatGPT.
>
> **Suggestion #4:** This is an old approach — why not use a generative approach? (line 120)
>
>
> **Response to #4:** We observed that the current studies of generative methods exclusively present ATE performance on the ASTE dataset, which is a subset of the complete SemEval dataset. They haven’t yet extended their evaluations to the entire ATE dataset. Consequently, we have not previously conducted comprehensive experiments on the generative methods. Your inquiries are highly valuable, and we are contemplating the incorporation of a generation approach.
>
>
> **Suggestion #5:** How about using S-BERT? (line 178)
>
>
> **Response to #5:** To simplify this work, we simply employ the embedding layer from a model fine-tuned on the ATE task. We will consider your question positively and explore using other models that focus on similarity recall (e.g. S-BERT) to further improve the accuracy of synonym replacement in our future work.
>
>
> **Suggestion #6:** How obtained? (line 207)
>
>
> **Response to #6:** When we employed ChatGPT to generate ten roles, we did not impose stringent restrictions. Consequently, certain roles may exhibit noise, potentially leading to adverse impacts on results. For instance, the purchasing manager may not be capable of providing valuable information regarding the taste of food, thereby introducing aspect terms unrelated to the review.
>
>
> We manually observed a relatively good positive effect of the voting rate at 0.7 and 0.8 based on a small dataset taken from the validation set. It is worth noting that an in-depth investigation into the roles’ reliability presents interesting challenges.
>
>
> **Suggestion #7:** Why are the results in Table 3 for BERT-pt-based PST without CHEF different from the results in Table 2 for Wang et al 2021? Isn’t this exactly the same setup? If not, what is the difference? The difference between Table 2 and Table 3 for BERT-pt-based PST is much larger than the improvement you show here from CHEF! (line 252)
>
>
> **Response to #7:** Table 2 shows the performance of PST reported in their paper. Table 3 shows the performance of our reproduced PST.
>
>
> We were unable to access the comprehensive augmenting data utilized by PST for training. Consequently, we recreated the augmented data by following the procedure outlined in their paper, which may have resulted in some deviations. To accurately reflect the performance improvement achieved by our approach, we carried out the comparison in Table 3.
>
>
> **Suggestion #8:** These improvements are tiny and the question is whether any of them are statistically significant? (line 254)
>
>
> **Response to #8:** We totally agree with your opinion that the improvement on some datasets is insignificant in amplitude. However, our approach simply uses ChatGPT as an auxiliary role without training the model. We will show the statistical significance analysis results in the Appendix and provide all the results in the extended version.
>
>
> **Suggestion #9:** You really should do ablation studies to show which part of CHEF helps (generating paraphrases; notion of perspective; synonym replacement; voting. The reader wants to know why CHEF is helping! (line 257)
>
>
> **Response to #9:** We have meticulously considered your suggestion, and present the results of ablation studies in the Appendix behind these replies. Besides, We intend to provide all the results in the extended version.
>
>
> **Suggestion #10:** This suggests precision is going down. You should always provide both recall and precision in your results so readers can understand what is happening. F1 is just one way of combing R and P, maybe some readers are interested in increasing R only for their problem and they care less about precision. (line 274)
>
>
> **Response to #10:** The provided recall value is derived from instances in which the model made errors in the original sentences. This is intended to demonstrate that our approach can effectively recall more aspect terms.
>
> We are willing to provide data reflecting changes in recall and precision across all the samples. It shows a decline in precision and a corresponding increase in recall, which could potentially be improved through further enhancements in our post-processing techniques. In Table 3 in the Appendix (behind our replies), we present an overview of the fluctuations in precision (P), recall (R), and F1 scores for our method when applied to BERT-base.
>
>
>
> ----------Appendix ----------
>
>
> **Table 1. Ablation Studies**
>
>
> | **Res14**                    | Restaurant Owner      | Chef                  | Waiter            | Restaurant Manager     | Purchasing Manager             | Catering Consultant      | Receptionist     | Financial Officer      | Cleaning Staff     | Diner                      |
> | ---------------------------- | --------------------- | --------------------- | ----------------- | ---------------------- | ------------------------------ | ------------------------ | ---------------- | ---------------------- | ------------------ | -------------------------- |
> | BERT-base                    | 88.03                 | 88.03                 | 88.03             | 88.03                  | 88.03                          | 88.03                    | 88.03            | 88.03                  | 88.03              | 88.03                      |
> | only combine (lines 117-118) | 70.65                 | 61.47                 | 68.82             | 65.51                  | 51.94                          | 60.87                    | 70.14            | 61.06                  | 64.66              | 68.11                      |
> | redundancy elimination       | 87.10                 | 87.67                 | 87.28             | 87.15                  | 86.60                          | 87.19                    | 87.60            | 87.27                  | 87.35              | 87.40                      |
> | synonym replacement          | 87.04                 | 87.45                 | 87.69             | 87.26                  | 87.54                          | 87.22                    | 87.80            | 87.69                  | 87.27              | 87.25                      |
> | multi-role voting            | 88.25                 | 88.25                 | 88.25             | 88.25                  | 88.25                          | 88.25                    | 88.25            | 88.25                  | 88.25              | 88.25                      |
> | **Res15**                    | **Restaurant Owner**  | **Chef**              | **Waiter**        | **Restaurant Manager** | **Purchasing Manager**         | **Catering Consultant**  | **Receptionist** | **Financial Officer**  | **Cleaning Staff** | **Diner**                  |
> | BERT-base                    | 72.17                 | 72.17                 | 72.17             | 72.17                  | 72.17                          | 72.17                    | 72.17            | 72.17                  | 72.17              | 72.17                      |
> | only combine (lines 117-118) | 57.24                 | 47.92                 | 55.64             | 53.48                  | 47.7                           | 49.65                    | 56.68            | 53.72                  | 45.66              | 50.68                      |
> | redundancy elimination       | 71.93                 | 70.59                 | 71.70             | 72.18                  | 70.80                          | 71.67                    | 71.64            | 71.55                  | 71.01              | 71.56                      |
> | synonym replacement          | 71.93                 | 71.00                 | 71.54             | 71.31                  | 70.85                          | 71.53                    | 71.51            | 70.78                  | 70.72              | 70.83                      |
> | multi-role voting            | 72.92                 | 72.92                 | 72.92             | 72.92                  | 72.92                          | 72.92                    | 72.92            | 72.92                  | 72.92              | 72.92                      |
> | **Res16**                    | **Restaurant Owner**  | **Chef**              | **Waiter**        | **Restaurant Manager** | **Purchasing Manager**         | **Catering Consultant**  | **Receptionist** | **Financial Officer**  | **Cleaning Staff** | **Diner**                  |
> | BERT-base                    | 74.88                 | 74.88                 | 74.88             | 74.88                  | 74.88                          | 74.88                    | 74.88            | 74.88                  | 74.88              | 74.88                      |
> | only combine (lines 117-118) | 61.55                 | 54.41                 | 62.66             | 58.75                  | 51.12                          | 55.58                    | 62.56            | 58.09                  | 56.03              | 57.14                      |
> | redundancy elimination       | 73.60                 | 74.32                 | 74.20             | 74.26                  | 73.98                          | 73.90                    | 73.72            | 73.50                  | 73.98              | 74.51                      |
> | synonym replacement          | 74.87                 | 74.04                 | 74.48             | 74.54                  | 74.53                          | 73.81                    | 73.47            | 72.89                  | 74.06              | 74.10                      |
> | multi-role voting            | 75.56                 | 75.56                 | 75.56             | 75.56                  | 75.56                          | 75.56                    | 75.56            | 75.56                  | 75.56              | 75.56                      |
> | **Lap14**                    | **Hardware Engineer** | **Software Engineer** | **Case Designer** | **Test Engineer**      | **Technical Support Engineer** | **Supply Chain Manager** | **Marketer**     | **Purchasing Manager** | **User**           | **Customer Service Staff** |
> | BERT-base                    | 82.96                 | 82.96                 | 82.96             | 82.96                  | 82.96                          | 82.96                    | 82.96            | 82.96                  | 82.96              | 82.96                      |
> | only combine (lines 117-118) | 42.99                 | 42.17                 | 40.93             | 51.41                  | 55.03                          | 45.54                    | 50.33            | 43.65                  | 62.26              | 58.34                      |
> | redundancy elimination       | 82.16                 | 82.32                 | 81.89             | 82.59                  | 81.71                          | 82.25                    | 82.88            | 82.33                  | 82.32              | 81.93                      |
> | synonym replacement          | 82.45                 | 82.13                 | 82.22             | 82.82                  | 82.18                          | 82.53                    | 82.84            | 82.83                  | 82.81              | 82.04                      |
> | multi-role voting            | 83.39                 | 83.39                 | 83.39             | 83.39                  | 83.39                          | 83.39                    | 83.39            | 83.39                  | 83.39              | 83.39                      |
>
>
>
> **Examples**
>
> **1. Example of redundancy elimination:**
>
> **Original**: Better than nobu, tomoe, taka, yama, sugiyama, and the rest.
>
> **Waiter**: Our **chef's sushi** is so amazing, it's better than Nobu, Tomoe, Taka, Yama, Sugiyama, and other well-known sushi spots in the city.
>
> **Ground-truth**: [No Aspect]
>
> **Output from the rewritten sentence**: chef's sushi
>
>
>
> **2. Example of synonym replacement**
>
> **Original**: Best **Indian food** I have ever eaten.
>
> **Chef**: I put a lot of effort into perfecting our **Indian dishes**.
>
> **Ground-truth**: Indian food
>
> **Output from the rewritten sentence**: Indian dishes
>
>
>
> **3. Example of multi-role voting**
>
> **Original**: I have to say they have one of the fastest **delivery times** in the city.
>
> **Chef**: I can prepare the order quickly to ensure timely **delivery**.
>
> **Ground-truth**: delivery times
>
> **Output from the rewritten sentence**: delivery
>
>
> **Table 2: Statistical Significance Analysis**
> | P-value                         | Lap14 | Res14 | Res15 | Res16 |
> | ------------------------------- | ----- | ----- | ----- | ----- |
> | Bert-base VS. BERT-base w/ CHEF | 0.009 | 0.046 | 0.002 | 0.021 |
> | PST VS. PST w/ CHEF             | 0.015 | 0.049 | -     | -     |
>
>
> **Table 3: Performance change**
> | %    | Lap14        | Res14        | Res15        | Res16        |
> | ---- | ------------ | ------------ | ------------ | ------------ |
> | P    | 85.97->84.98 | 88.23->86.82 | 70.28->70.07 | 74.76->74.37 |
> | R    | 80.15->81.85 | 88.85->89.73 | 74.17->76.01 | 75.00->76.80 |
> | F    | 82.96->83.39 | 88.03->88.25 | 72.17->72.92 | 74.88->75.56 |
>
> Due to the long training time of PST, accurate statistical significance analysis results could not be obtained in the short term. We intend to provide all the results in the extended version.
>
>
> Thank you again for your insightful feedback, which undoubtedly contributes to enhancing the quality and enriching the final version of the paper.

---

### Official Review · Reviewer_1MTq · 2023-08-06

**Soundness:** 3

**Excitement:**

3: Ambivalent: It has merits (e.g., it reports state-of-the-art results, the idea is nice), but there are key weaknesses (e.g., it describes incremental work), and it can significantly benefit from another round of revision. However, I won't object to accepting it if my co-reviewers champion it.

**Paper Topic And Main Contributions:**

This paper presents a way to utilize ChatGPT to help pre-existing PLM-based tools for Aspect Term Extraction (ATE), through relevant-role-prompted rewriting of input sentences. This may be seen as using domain-specific prompt-engineering to augment input sentences, implemented with ChatGPT and empirical performance evaluated on four groups of sentences covering two domains (Laptops and Restaurants) from SemEval datasets.

**Questions For The Authors:**

A. How sensitive are the empirical results from perturbing the multi-role voting related hyperparameter values? Especially, the overall empirical results carry some level of "still do no harm in worst case" argument, and I am curious whether this is an empirically robust claim.

**Reasons To Accept:**

It is a readily-applicable technique that leverages ChatGPT, and it can be applied to a broad class of methods to solve ATE tasks. The results in Table 2 seem to be a solid incremental result over previously published techniques. Straightforward idea, easy-to-use choice of tool, tangible improvements and potentially broad applicability make this paper a reasonable candidate.

**Reasons To Reject:**

Empirical validation covers a narrow set of domains, and the reported improvements in Table 3 and Figure 2 are mostly positive but can be seen as finely tuned results. In addition to the hyperparameters explicitly discussed in section 4.2, the voting section (lines 201-208) seems to contain more hyperparameters whose sensitivity to the final result is not presented.


**Reproducibility:**

4: Could mostly reproduce the results, but there may be some variation because of sample variance or minor variations in their interpretation of the protocol or method.

**Reviewer Confidence:**

3: Pretty sure, but there's a chance I missed something. Although I have a good feel for this area in general, I did not carefully check the paper's details, e.g., the math, experimental design, or novelty.

---

> ### Author Rebuttal · Authors · 2023-08-28
>
> We extend our sincere appreciation for your dedicated effort in reviewing our papers and offering valuable suggestions to enhance their quality. Your feedback holds significant value for us, and we genuinely consider each suggestion you provide. We take your feedback seriously, and our response is as follows:
>
>
>
> ----------Response to Comments----------
>
>
> **Comment #1:** Empirical validation covers a narrow set of domains, and the reported improvements in Table 3 and Figure 2 are mostly positive but can be seen as finely tuned results. In addition to the hyperparameters explicitly discussed in section 4.2, the voting section (lines 201-208) seems to contain more hyperparameters whose sensitivity to the final result is not presented.
>
>
> **Response to #1:** Our research is centered on the Aspect Term Extraction (ATE) task, specifically involving reviews as input without any additional features or sentiment analysis considerations. To facilitate a comprehensive comparison with prior ATE studies, our primary emphasis lies on the four SemEval datasets, which cover only these two domains.
>
>
> Our investigation revealed that the sentences rewritten by ChatGPT occasionally introduced noises. For example, the results extracted from the rewritten sentences may contain some aspect terms that did not exist in the original text, although such terms are appropriate according to our verification. Since our evaluation was grounded in the ground-truth of the original text, we found it necessary to implement filtering. These adjustments were made to ensure a fair and equitable comparison with the work of others.
>
> There is only one hyperparameter that needs to be set, which is the threshold limiting the number of different roles of virtual experts (the ones generated by ChatGPT) for voting. We used ChatGPT to automatically generate 10 virtual experts who have different roles in a specific domain, and allow 7 experts to join the voting process. This is an empirical setting according to the performance of the experts on a much smaller subset of data sampled from the validation set.
>
>
>
> ----------Response to Questions----------
>
>
> **Question #1:** How sensitive are the empirical results from perturbing the multi-role voting related hyperparameter values? Especially, the overall empirical results carry some level of "still do no harm in worst case" argument, and I am curious whether this is an empirically robust claim.
>
>
> **Response to #1:** You raise a very important issue, the acceptance rate (line 200) is very sensitive to the final result.
>
>
> We employed ChatGPT to generate ten experts without imposing stringent constraints. It's worth noting that, in certain instances, this approach may introduce noise, potentially leading to adverse effects on sentences. For instance, the purchasing manager might not be capable of providing valuable information regarding food taste. Consequently, we have devised an integrated system that employs a voting mechanism to address these concerns.
>
>
> We manually observed a relatively positive effect of vote rate at 0.7 and 0.8 based on a small subset of data sampled from the validation set. We appreciate that an in-depth investigation into the roles' reliability presents interesting challenges.
>
>
> We sincerely appreciate the constructive critique provided by you, which has undeniably enriched the final version of the paper. Thank you for your time and consideration.

---

### Official Review · Reviewer_m3uM · 2023-08-11

**Soundness:** 3

**Excitement:**

4: Strong: This paper deepens the understanding of some phenomenon or lowers the barriers to an existing research direction.

**Paper Topic And Main Contributions:**

The paper focuses on the aspect triplet extraction task and proposes a ChatGPT-based Edition Fictionalization (CHEF) method to assist the current PLM-based extractors.

**Reasons To Accept:**

1. The motivation of the paper is clear, sensible and evident to the proposed method.
2. The proposed idea is overall novel and interesting, which effectively emploits an LLM to find experts and generate more high-quality data to facilitate the extractor.
3. The writing is generally good, which I can easily follow. The examples displayed in tables or raw text makes the illustration very clear and easy to understand.

**Reasons To Reject:**

1. The paper claims that CHEF can paraphrase the original sentence into a more informative and high-quality one, but during the process it also injects model bias from Chat-GPT into the generated sentence, which will shift the distribution of training data away from test data.  This kind of noises can counteract the benefits from more wonderful data, and as it shows in Table 3 the improvements is not very significant probably due to this.

2. The paper lacks an analysis or statement about the time/financial cost of the methods (e.g., the cost of Chat-GPT inference in terms of USD).

**Reproducibility:**

3: Could reproduce the results with some difficulty. The settings of parameters are underspecified or subjectively determined; the training/evaluation data are not widely available.

**Reviewer Confidence:**

3: Pretty sure, but there's a chance I missed something. Although I have a good feel for this area in general, I did not carefully check the paper's details, e.g., the math, experimental design, or novelty.

---

> ### Author Rebuttal · Authors · 2023-08-28
>
> We would like to express our heartfelt appreciation for dedicating your valuable time to reviewing our paper and offering us invaluable insights and suggestions aimed at enhancing its quality. Your questions have made us aware of the key to further improving the paper We have taken careful consideration of each of your suggestions and comments, and we hereby provide our responses to each point as below.
>
>
>
> ----------Response to Comments----------
>
>
> **Comment #1:** The paper claims that CHEF can paraphrase the original sentence into a more informative and high-quality one, but during the process it also injects model bias from Chat-GPT into the generated sentence, which will shift the distribution of training data away from test data. This kind of noises can counteract the benefits from more wonderful data, and as it shows in Table 3 the improvements is not very significant probably due to this.
>
>
> **Response to #1:** We totally recognize your point of view. To address noises, we have introduced post-processing operations, including redundancy elimination, synonym replacement, and multi-role voting.
>
> Our method exclusively combines the extracted results from both the original and rewritten sentences and performs post-processing. It refrains from retraining the ATE model, thus avoiding any potential distribution shifting away from the test data. All our post-processing operations are straightforward to implement without retraining.
>
>
> **Comment #2:** The paper lacks an analysis or statement about the time/financial cost of the methods (e.g., the cost of Chat-GPT inference in terms of USD).
>
>
> **Response to #2:** Our method obviates the necessity for model retraining, requiring solely the rewriting of sentences within the test set, encompassing a total of 2,963 sentences. The request of ChatGPT for generating ten rewritten sentences for each test sample yielded an average response time of 2.5 seconds, resulting in an approximate total cost of $90. In forthcoming research, we plan to leverage ChatGPT to distill a LLaMA model for rewriting sentences, thereby mitigating API expenses.
>
>
>
> We extend our heartfelt gratitude for your insightful comments, which will help us improve the quality of our paper.

---

### Official Review · Reviewer_p6WA · 2023-08-18

**Soundness:** 3

**Excitement:**

4: Strong: This paper deepens the understanding of some phenomenon or lowers the barriers to an existing research direction.

**Paper Topic And Main Contributions:**

This paper aims to improve aspect term extraction by performing automatic rewriting tasks for domain-specific aspect extraction using the lens of subject experts using ChatGPT. The methodology is tested on the SemEval task corpora L14 and R14-16 and shows an increase in F1-score and performance. The method, known as ChatGPT-based Edition Fictionalization, does not pretrain or fine-tune ChatGPT in this rewriting exercise with a list of virtual experts which are generated by ChatGPT using prompts, following which the method uses the generated list of roles as keys to rewrite the input sentence for classification.

**Questions For The Authors:**

1. What other groups of prompts were tried for this purpose?
2. Does the model performance scale beyond these two domains?
3. In the limitations section, there is a claim that "the proposed method yields slight improvement when coupled with domain-specific models." Could you provide some qualitative examples to highlight this difference along with the domain-specific models in question?


**Reasons To Accept:**

1. This is a novel method of using ChatGPT prompt engineering for a classification task
2. The method seeks to be more controllable than traditional LLM-based classification methods, as the prompts used by ChatGPT are human-readable and editable, and the post-processing looks traditional and rule-based (including synonym replacement and multi-role voting for contentious results)
3. The performance of the models shows promise that controlled prompting of ChatGPT-like models can help improve model performance across select classification tasks

**Reasons To Reject:**

1. A more extensive comparison across models, including the current aspect extraction state-of-the-art as well as a detailed error analysis of this pipeline would have been useful.

**Reproducibility:**

4: Could mostly reproduce the results, but there may be some variation because of sample variance or minor variations in their interpretation of the protocol or method.

**Reviewer Confidence:**

4: Quite sure. I tried to check the important points carefully. It's unlikely, though conceivable, that I missed something that should affect my ratings.

---

> ### Author Rebuttal · Authors · 2023-08-28
>
> We extend our sincere gratitude for dedicating your valuable time to reviewing our paper and providing us with insightful questions and suggestions. We have meticulously considered each of your suggestions and comments. We answer the concerns as below:
>
>
>
> ----------Response to Comments----------
>
>
> **Comment #1:** A more extensive comparison across models, including the current aspect extraction state-of-the-art as well as a detailed error analysis of this pipeline would have been useful.
>
>
> **Response to #1:** We conducted redundancy elimination, synonym replacement, and multi-role voting on the extracted results. These auxiliary techniques were implemented to ensure the overall effectiveness of our approach and the fairness of comparison with other methods.
>
> We realize that the mentioned error analysis aligns with the ablation study, where the synonym replacement and multi-role voting need to be ablated progressively for insight into the performance in different phrases. We provide a detailed ablation study for every generated role in Table 1 in the Appendix behind our replies. Besides, we provide examples to assist the observation.
>
> Due to the page limit, we haven’t provided exhaustive contents. In the extended version, we will be delighted to present our more comprehensive experimental data.
>
>
>
> ----------Response to Questions----------
>
>
> **Question #1:** What other groups of prompts were tried for this purpose?
>
>
> **Response to #1:** We didn't try other prompts. The sole prompt we tried at the very beginning works well. In other words, we consistently utilized a single prompt without employing techniques such as context learning, which made it easy to follow.
>
>
> **Question #2:** Does the model performance scale beyond these two domains?
>
>
> **Response to #2:** At present, our focus is restricted solely to the ABSA datasets from SemEval for Aspect Term Extraction (ATE) research. The datasets include only the restaurant and laptop domains.
>
> We also realized other existing datasets that are relevant to ATE, such as the COQE task that involves the camera and car domains. However, we concentrate on pure ATE task research without any additional information interventions.
>
> We are considering the application of this approach in future endeavors, specifically for tasks such as ABSA and COQE, in order to assess its reliability within an end-to-end system. During this phase, we can explore potential applications in other domains.
>
>
> **Question #3:** In the limitations section, there is a claim that "the proposed method yields slight improvement when coupled with domain-specific models." Could you provide some qualitative examples to highlight this difference along with the domain-specific models in question?
>
> **Response #3:** We are so sorry that the section of limitations may have been misleading. We want to convey that our approach can yield performance improvements on a variety of "domain-specific models", where we mean those trained on a specific domain of the ATE task. In our experiments, we employed three models: BERT-base, PST (both detailed in the paper), and one that we are currently working on. Our approach demonstrated consistent improvements across all these models. We are enthusiastic about sharing the data in the final revision.
>
>
>
> ----------Appendix----------
>
>
> **Table 1. Ablation Studies for Different Roles**
> | **Res14**                    | Restaurant Owner  | Chef              | Waiter        | Restaurant Manager | Purchasing Manager         | Catering Consultant  | Receptionist | Financial Officer  | Cleaning Staff | Diner                  |
> | ---------------------------- | ----------------- | ----------------- | ------------- | ------------------ | -------------------------- | -------------------- | ------------ | ------------------ | -------------- | ---------------------- |
> | BERT-base                    | 88.03             | 88.03             | 88.03         | 88.03              | 88.03                      | 88.03                | 88.03        | 88.03              | 88.03          | 88.03                  |
> | only combine (lines 117-118) | 70.65             | 61.47             | 68.82         | 65.51              | 51.94                      | 60.87                | 70.14        | 61.06              | 64.66          | 68.11                  |
> | redundancy elimination       | 87.10             | 87.67             | 87.28         | 87.15              | 86.60                      | 87.19                | 87.60        | 87.27              | 87.35          | 87.40                  |
> | synonym replacement          | 87.04             | 87.45             | 87.69         | 87.26              | 87.54                      | 87.22                | 87.80        | 87.69              | 87.27          | 87.25                  |
> | multi-role voting            | 88.25             | 88.25             | 88.25         | 88.25              | 88.25                      | 88.25                | 88.25        | 88.25              | 88.25          | 88.25                  |
> | **Res15**                    | **Restaurant Owner**  | **Chef**              | **Waiter**       | **Restaurant Manager** | **Purchasing Manager**         | **Catering Consultant**  | **Receptionist** | **Financial Officer**  | **Cleaning Staff** | **Diner**                  |
> | BERT-base                    | 72.17             | 72.17             | 72.17         | 72.17              | 72.17                      | 72.17                | 72.17        | 72.17              | 72.17          | 72.17                  |
> | only combine (lines 117-118) | 57.24             | 47.92             | 55.64         | 53.48              | 47.7                       | 49.65                | 56.68        | 53.72              | 45.66          | 50.68                  |
> | redundancy elimination       | 71.93             | 70.59             | 71.70         | 72.18              | 70.80                      | 71.67                | 71.64        | 71.55              | 71.01          | 71.56                  |
> | synonym replacement          | 71.93             | 71.00             | 71.54         | 71.31              | 70.85                      | 71.53                | 71.51        | 70.78              | 70.72          | 70.83                  |
> | multi-role voting            | 72.92             | 72.92             | 72.92         | 72.92              | 72.92                      | 72.92                | 72.92        | 72.92              | 72.92          | 72.92                  |
> | **Res16**                    | **Restaurant Owner**  | **Chef**              | **Waiter**       | **Restaurant Manager** | **Purchasing Manager**         | **Catering Consultant**  | **Receptionist** | **Financial Officer**  | **Cleaning Staff** | **Diner**                  |
> | BERT-base                    | 74.88             | 74.88             | 74.88         | 74.88              | 74.88                      | 74.88                | 74.88        | 74.88              | 74.88          | 74.88                  |
> | only combine (lines 117-118) | 61.55             | 54.41             | 62.66         | 58.75              | 51.12                      | 55.58                | 62.56        | 58.09              | 56.03          | 57.14                  |
> | redundancy elimination       | 73.60             | 74.32             | 74.20         | 74.26              | 73.98                      | 73.90                | 73.72        | 73.50              | 73.98          | 74.51                  |
> | synonym replacement          | 74.87             | 74.04             | 74.48         | 74.54              | 74.53                      | 73.81                | 73.47        | 72.89              | 74.06          | 74.10                  |
> | multi-role voting            | 75.56             | 75.56             | 75.56         | 75.56              | 75.56                      | 75.56                | 75.56        | 75.56              | 75.56          | 75.56                  |
> | **Lap14**                    | **Hardware Engineer** | **Software Engineer** | **Case Designer** | **Test Engineer**      | **Technical Support Engineer** | **Supply Chain Manager** | **Marketer**     | **Purchasing Manager** | **User**           | **Customer Service Staff** |
> | BERT-base                    | 82.96             | 82.96             | 82.96         | 82.96              | 82.96                      | 82.96                | 82.96        | 82.96              | 82.96          | 82.96                  |
> | only combine (lines 117-118) | 42.99             | 42.17             | 40.93         | 51.41              | 55.03                      | 45.54                | 50.33        | 43.65              | 62.26          | 58.34                  |
> | redundancy elimination       | 82.16             | 82.32             | 81.89         | 82.59              | 81.71                      | 82.25                | 82.88        | 82.33              | 82.32          | 81.93                  |
> | synonym replacement          | 82.45             | 82.13             | 82.22         | 82.82              | 82.18                      | 82.53                | 82.84        | 82.83              | 82.81          | 82.04                  |
> | multi-role voting            | 83.39             | 83.39             | 83.39         | 83.39              | 83.39                      | 83.39                | 83.39        | 83.39              | 83.39          | 83.39                  |
>
>
>
> **Examples**
>
> **1. Example of redundancy elimination:**
>
> **Original**: Better than nobu, tomoe, taka, yama, sugiyama, and the rest.
>
> **Waiter**: Our **chef's sushi** is so amazing, it's better than Nobu, Tomoe, Taka, Yama, Sugiyama, and other well-known sushi spots in the city.
>
> **Ground-truth**: [No Aspect]
>
> **Output from the rewritten sentence**: chef's sushi
>
>
>
> **2. Example of synonym replacement**
>
> **Original**: Best **Indian food** I have ever eaten.
>
> **Chef**: I put a lot of effort into perfecting our **Indian dishes**.
>
> **Ground-truth**: Indian food
>
> **Output from the rewritten sentence**: Indian dishes
>
>
>
> **3. Example of multi-role voting**
>
> **Original**: I have to say they have one of the fastest **delivery times** in the city.
>
> **Chef**: I can prepare the order quickly to ensure timely **delivery**.
>
> **Ground-truth**: delivery times
>
> **Output from the rewritten sentence**: delivery
>
>
>
> We are grateful for your insightful comments, which undoubtedly strengthen the quality of our paper. Your guidance has played a crucial role in refining our work, making it more coherent and impactful.

---

### Meta-Review · Area_Chair_hc3g · 2023-09-19

**Recommendation:** 3

**Metareview:**

This paper explores a method for aspect term extraction that leverages ChatGPT to generates higher quality data to facilitate the predictions. The reviewers have several concerns for the paper regarding marginal performance improvement and additional experiments for the contribution of different steps in the pipeline. The rebuttal has provided further experiment results and clarified some details. The authors are encouraged to update the paper according to the rebuttal and discussion with the reviewers.

---

### Decision · Program_Chairs · 2023-10-07

**Decision:**

Accept-Findings

**Comment:**

This paper explores a method for aspect term extraction that leverages ChatGPT to generates higher quality data to facilitate the predictions. The reviewers have several concerns for the paper regarding marginal performance improvement and additional experiments for the contribution of different steps in the pipeline. The rebuttal has provided further experiment results and clarified some details. The authors are encouraged to update the paper according to the rebuttal and discussion with the reviewers.